# Laser-Induced Breakdown Spectroscopy Analysis of Lithium: A Comprehensive Review

**DOI:** 10.3390/s25247689

**Published:** 2025-12-18

**Authors:** Stefano Legnaioli, Giulia Lorenzetti, Francesco Poggialini, Beatrice Campanella, Vincenzo Palleschi, Silvana De Iuliis, Laura Eleonora Depero, Laura Borgese, Elza Bontempi, Simona Raneri

**Affiliations:** 1Institute of Chemistry of Organometallic Compounds, National Research Council, 56124 Pisa, Italy; stefano.legnaioli@cnr.it (S.L.); giulia.lorenzetti@cnr.it (G.L.); francesco.poggialini@cnr.it (F.P.); beatrice.campanella@cnr.it (B.C.); 2Institute of Condensed Matter Chemistry and Energy Technologies, National Research Council, 20125 Milano, Italy; silvana.deiuliis@cnr.it; 3INSTM and Chemistry for Technologies Laboratory, Department of Mechanical and Industrial Engineering, University of Brescia, 25121 Brescia, Italy; laura.depero@unibs.it (L.E.D.); laura.borgese@unibs.it (L.B.); elza.bontempi@unibs.it (E.B.); 4Department of Earth Sciences, University of Florence, 50121 Florence, Italy; simona.raneri@unifi.it

**Keywords:** lithium, LIBS, lithium-ion batteries, recycling, circular economy

## Abstract

Lithium has emerged as a pivotal material for the global energy transition, yet its supply security is challenged by limited geographical availability and growing demand. These constraints highlight the need for analytical methods that are not only accurate but also sustainable and deployable across the entire lithium value chain. In this context, Laser-Induced Breakdown Spectroscopy (LIBS) offers distinctive advantages, including minimal sample preparation, real-time and in situ analysis and the potential for portable and automated implementation. Such features make LIBS a valuable tool for monitoring and optimizing lithium extraction, refining and recycling processes. This review critically examines the recent progress in the use of LIBS for lithium detection and quantification in geological, industrial, biological and extraterrestrial matrices. It also discusses emerging applications in closed-loop recycling and highlights future prospects related to the integration of LIBS with artificial intelligence and machine learning to enhance analytical accuracy and material classification.

## 1. Introduction

Lithium (Li) is a lightweight, silvery-white alkali metal with atomic number 3. It is present in nature in two stable isotopic forms, ^6^Li (about 7.5%) and ^7^Li (about 92.5%). The Grotrian diagram of Li I is shown in Figure 1.

The relative simplicity of the electron configuration of lithium is reflected in the presence of only a few emission lines in its optical spectrum, which are reported in Table 1 together with the corresponding energies E_i_ and E_k_ of the upper and lower levels of the transition, the degeneracies of the levels g*i* and g_k_ and the transition probability A_ki_.

It is worth noting that the two intense lithium lines at 670.78 nm and 610.36 nm are characterized by very different upper-level energies (1.85 eV and 3.88 eV, respectively). Therefore, the choice of one of these lines for analytical studies depends on the plasma electron temperature, since the intensity of the line is proportional to the population of the upper level of the transition, which in turn depends exponentially on the ratio between the energy of the level and the thermal energy k_B_T [1].

**Table 1 sensors-25-07689-t001:** Main emission lines of Li I in the optical range. Data from the NIST Atomic database [2].

Wavelength (nm)	Transition	E_i_ (eV)	E_k_ (eV)	g_i_	g_k_	A_ki_ (s^−1^)
670.78	1*s*^2^2p-1*s*^2^2s	0	1.85	2	6	3.69 × 10^7^
323.27	1*s*^2^3p-1*s*^2^2s	0	3.83	2	6	1.00 × 10^6^
274.12	1*s*^2^4p-1*s*^2^2s	0	4.52	2	6	1.25 × 10^6^
812.64	1*s*^2^3s-1*s*^2^2p	1.85	3.37	6	2	3.35 × 10^7^
610.36	1*s*^2^3d-1*s*^2^2p	1.85	3.88	6	10	6.86 × 10^7^
497.17	1*s*^2^4s-1*s*^2^2p	1.85	4.34	6	2	1.04 × 10^7^
460.29	1*s*^2^4d-1*s*^2^2p	1.85	4.54	6	10	2.32 × 10^7^

Lithium has become essential to the advancement of modern technologies as the key element of rechargeable lithium-ion batteries used in electric vehicles (EVs) and storage systems for power grids, as well as consumer electronics. The rapid growth of electric mobility and renewable energy storage has exponentially amplified the global demand for lithium resources, whose exploitation is nowadays mainly based on natural resources, namely brine and rock ores. Additionally, the European Commission has classified it as a critical raw material and also “strategic” within the framework of the Critical Raw Materials Act [3], since it is considered essential to achieve the EU’s climate and digital goals and because its supply chains are concentrated in a few countries [4].

Applications considered critical include, in particular, batteries for electric mobility, stationary storage for renewable sources and, more generally, energy and digital technologies of industrial interest [5].

The use of lithium, however, is not limited to the energy storage market. Its distinctive physicochemical characteristics make it a fundamental component in various industrial sectors, such as the ceramics and glass industries, lubricating greases, air treatment, continuous casting mold flux powders and medical uses [6,7] (see Figure 2).

Natural deposits of lithium consist in granitic pegmatites and *salares*. In pegmatite rocks, lithium is found primarily in spodumene (LiAlSi_2_O_6_), a lithium and aluminum silicate that typically contains about 3.7 wt% of Li. Other important lithium-bearing minerals include lepidolite KLi_1.5_Al_1.5_(Si_3_O_10_)(F,OH)_2_, petalite LiAlSi_4_O_10_ and zinnwaldite KLiFeAl(AlSi_3_)O_10_(OH,F)_2_. Major pegmatite deposits are located in Australia, Canada, Portugal, Brazil and Zimbabwe.

Another large contribution to the global lithium production comes from a region encompassing the borders of Argentina, Chile and Bolivia, called the ‘lithium triangle’. In this region lithium is extracted from *salares* (salt pans) (Figure 3); the Salar de Atacama, in Chile, has the highest Li concentration in brine, of the order of 0.15% in weight. Chile is the largest producer of lithium in the region, largely surpassing Argentina and Bolivia, in that order. In the East, China is the third largest producer of lithium, which is extracted from rocks and brine (see Table 2). On the other hand, China is particularly active in lithium manufacturing, mostly for the production of lithium batteries [8].

Recently, interest in and the importance of volcano-sedimentary deposits has increased. In these deposits, lithium occurs within the structure of clay minerals. They currently represent the third largest natural source of lithium worldwide, contributing approximately 8% of total production. Significant deposits of this type are found in the western United States (e.g., the McDermitt Caldera and King Valley, along the Nevada–Oregon border) and in southern China. In Jadar, Serbia, there is a unique deposit, probably the largest in Europe, where the main lithium-bearing mineral is jadarite, LiNaSiB_3_O_7_(OH).

According to some analysts, the world demand for lithium will grow by a factor ranging from 3.5 to 4.2 in the next decade, mostly driven by the electric car market and renewable energy storage systems [9].

On the other hand, the geopolitical issues related to the dominating role of China in lithium manufacturing and the nationalistic policy of lithium-producing countries in Latin America, combined with the ever-growing concerns about the very high environmental impact associated with the recovery of lithium from the *salares*, along with the many other variables that might influence the markets, are pushing several countries, including the ones belonging to the European Union, to diversify supplies, develop refining and recycling capacities, find alternative sources, explore extraction from secondary raw materials and support strategic projects along the battery value chain so as to ensure access to the lithium needed for electric mobility and storage.

The lithium content within these resources can vary significantly. Therefore, it is essential to determine the lithium concentration prior to their exploitation. Consequently, the availability of reliable and efficient techniques to assess lithium occurrence both in geological formations and in alternative sources is of great importance. This review provides a comprehensive overview of LIBS applications for lithium detection and quantification across different matrices, summarizing recent advances and outlining future research directions.

## 2. Techniques for Lithium Analysis

Fast and reliable analytical methods are essential to ensure the traceability and sustainability of the lithium supply chain, from monitoring extraction and refining processes to ensuring quality control during the recycling of lithium-ion batteries. A recent review by Chaudry et al. [10] discusses several of the methods currently used for lithium analysis; similarly, Workman [11] reviewed current spectroscopic techniques for the analysis of lithium-ion batteries.

At present, the analysis of lithium-containing samples is carried out primarily in laboratories using analytical instrumentation such as inductively coupled plasma optical emission spectroscopy (ICP-OES) [12,13], inductively coupled plasma mass spectrometry (ICP-MS) [14,15], atomic absorption spectrometry (AAS) [16,17], ion chromatography (IC) [18,19], spark optical emission spectroscopy (Spark-OES) [20,21], X-ray fluorescence (XRF) [22,23], among others.

The analytical characteristics of these techniques are compared with those of LIBS in Table 3, and the comparison is illustrated graphically in Figure 4. Among the analytical factors of merit, particularly important is the Limit of Detection (*LOD*), which corresponds to the minimum concentration of the analyte that can be detected using the technique with a reasonable probability. The formula usually applied for the *LOD* is given by the following relation:(1)$$LOD=\frac{3\sigma }{b}$$
where *σ* is the standard deviation of the signal measured on a blank (i.e., at zero analyte concentration) and *b* is the slope of the calibration curve. It should be noted that the application of this definition in LIBS analysis has been recently criticized [24]; in fact, the definition (1) should be considered a lower limit for the *LOD*.

Another important analytical indicator is the Root Mean Square Error (*RMSE*), which is associated with the analytical precision of the technique. It is defined as follows:(2)$$RMSE=\sqrt{\frac{\sum_{i=1}^{n}{\left(p_{i}-r_{i}\right)}^{2}}{n}}$$
where n is the number of samples, $p_{i}$ are the predictions obtained from the experiment and $r_{i}$ are the corresponding reference values.

The authors highlighted the established traditional and modern methods for lithium analytical determination. However, they typically involve laborious sample preparation, are time-intensive and are unsuitable for real-time or in situ measurements. In addition, most of these analytical techniques are associated with certain drawbacks and challenges that hinder accurate lithium quantification. In particular, the presence of ions such as Na^+^, K^+^, Ca^2+^ and Mg^2+^ can significantly affect lithium detection by AAS, as increasing concentrations of these species tend to suppress the lithium signal. Similarly, in ICP-MS measurements, elements like sodium and potassium may interfere by reducing lithium ionization and causing spectral scattering or line overlap. To account for such interferences, the addition of an internal standard is often recommended when analyzing lithium in biological fluids. However, Na interference poses critical issues in lithium determination. Although ion cation-exchange (IC) can be used to separate lithium, residual Na remains a major obstacle, as it overlaps with lithium signals and necessitates a higher ionization filament current to achieve a stable Li^+^ beam intensity [25]. Such limitations have motivated the exploration of alternative analytical approaches offering rapid, accurate and non-destructive detection capabilities.

An emerging elemental technique known as Laser-Induced Breakdown Spectroscopy (LIBS) offers several potential advantages.

Laser-Induced Breakdown Spectroscopy is a laser-based multielement analytical technique whose origins date back to the early 1960s [26], shortly after the invention of the laser. However, the term LIBS became established only after the publication of two papers in 1981 by Tim Loree and Leon Radziemski [27,28] from the Los Alamos Laboratories in New Mexico, USA, who introduced the acronym for the first time. Leon Radziemski, together with his younger colleague David Cremers, further developed the technique in the United States up to the 1990s, when LIBS was first introduced in Europe by the Pisa group [29,30]. Since then, LIBS has evolved into a mature analytical technique, with research groups operating on the five continents. Their work has introduced its use in numerous applications, including industrial process monitoring [31], biomedical analysis [32], environmental protection [33], geochemistry [34,35], cultural heritage conservation and study [36,37], among many others.

The main feature of LIBS is the use of a single tool, the laser, for both sampling the material through the mechanism of laser ablation (on solid samples) or optical dielectric breakdown (on liquid or gaseous samples) and simultaneously atomizing and exciting the sampled material, bringing it into the plasma state [38]. The optical emission from the hot and dense laser-induced plasma is then collected and spectrally analyzed to determine the composition of the material. The use of a single tool for sampling and excitation is not unique to LIBS. For example, in Spark-Discharge Optical Emission Spectroscopy (SD-OES), ablation and excitation both occur through an electrical discharge; however, in LIBS, the use of a laser permits remote analysis of conducting and non-conducting materials, even without physical contact with the object under study. Moreover, LIBS analysis does not require sample preparation, a feature that, combined with the simplicity of the experimental setup and the minimal measurement time, makes the technique particularly suitable for in situ analysis [39,40,41,42]. All the above characteristics have also contributed to defining LIBS as a ‘green’ technique [43].

The recent development of hand-held LIBS instruments has further expanded the range of applications, especially in geological prospecting, industrial diagnostics and waste sorting and recycling [44,45,46,47,48]. In recent years, the analytical capabilities of the LIBS techniques have been further enhanced by the introduction of powerful Machine Learning (ML) algorithms, which allow for a fast and deeper analysis of the large amount of data that can be accumulated in LIBS experiments in a very short time [49,50].

## 3. LIBS for Lithium Analysis

The portability, rapidity and ease of use of the LIBS instrumentation have attracted considerable attention in all the analytical applications involving lithium analysis, especially the ones that could be performed in situ with portable instrumentation.

The interest in using LIBS for lithium analysis is further motivated by the fact that its stronger competitor in in situ applications, i.e., the Energy Dispersive X-Ray Fluorescence (ED-XRF) technique, cannot be used for the analysis of light elements, including lithium, which is the third element in the periodic table after hydrogen and helium.

A search on Scopus™ with keywords “laser AND induced AND breakdown AND spectroscopy AND lithium” reports 274 publications (225 articles, 42 conference papers, 5 reviews, 1 book chapter and 1 note) in the time range between 1999 and the time of writing this review (November 2025).

Note that searching for “LIBS AND lithium” would report more than 20,000 results, since LIBs is also the acronym for “Lithium-ion batteries”.

The number of publications shown in Figure 5 evidences a quick growth, from a few papers per year before 2010 to more than 30 in 2025. In the same period, the percentage of LIBS papers on lithium with respect to the global LIBS production has grown from 0.3% in 2010 to a solid 4% in 2025.

The most active country in LIBS research for lithium is China, with a total of 68 papers, followed by USA with 57 and Germany with 30 (see Figure 6).

From a historical point of view, the first paper on LIBS mentioning lithium was published in 1999 [51] in the SPIE Conference Proceedings by the group of David Hahn at Sandia National Laboratory in Livermore CA, USA. The topic was the development of optical sensors for process control and emission monitoring in industry. In that communication, among other sensors, a LIBS instrument was presented for hazardous metal detection in process streams.

In the period between 2010 and 2025, the research on the LIBS analysis of lithium moved along several main directions: analysis of lithium in rocks, analysis of lithium in liquids, applications to lithium recycling and recovery, industrial applications of lithium and analysis of LIBS in biological systems. These main topics will be presented in the next sections through the discussion of the relevant papers published in the last 15 years.

### 3.1. Rocks

Laser-Induced Breakdown Spectroscopy (LIBS) has proven highly effective for detecting lithium across a wide variety of geological matrices, including rocks, minerals and soils. Its ability to analyze both major and trace components in heterogeneous samples makes it a versatile tool for geoscientific investigations.

The technique is especially valued in geological studies due to its strong sensitivity to light elements, such as lithium, which are often difficult to measure reliably using other spectroscopic methods. LIBS can detect lithium even at low concentrations, providing researchers with important information about mineralogical composition, alteration processes and geochemical pathways. This sensitivity is crucial for applications ranging from mineral exploration to environmental monitoring of lithium-bearing sediments.

In addition to its analytical capabilities, LIBS is appreciated for its overall simplicity and operational flexibility. Instruments can be deployed in laboratory settings, mounted on field-portable systems, or even integrated into remote or autonomous platforms such as robotic rovers. This adaptability allows geologists and planetary scientists to collect elemental data rapidly and efficiently, making LIBS a powerful technique for both terrestrial and extraterrestrial geological applications.

The first reported use of LIBS for the analysis of lithium in geological applications was published in 2002 by Fabre et al. [52]. The authors performed an exploratory study on the use of LIBS for the analysis of lithium in solids. The recent diffusion of LIBS hand-held instrumentation has further widened the possible applications of the technique for in-field geological analysis [46]. The same group published several papers on the use of hand-held LIBS instrumentation for lithium analysis [23,40,53,54].

In 2013 the group of Pavel Veis [55] analyzed an acid pitchstone sample from Iceland, detecting in the LIBS spectra the presence of several elements, including Li. For determining the sample’s composition the authors used the Calibration-Free LIBS method.

CF-LIBS was introduced by the Pisa group in 1999 [56]. Since then, the technique has been improved, relaxing some of the original constraints [57,58], up to becoming nowadays a viable alternative to univariate or multivariate calibration methods when matrix-matched standards are not easily available. In [55] the authors declared the accuracy of the Li concentration estimated by CF-LIBS acceptable, considering the technique viable for geological purposes.

Ribeiro et al. [59] studied Li-rich minerals from the Argemela tin mine in Portugal using XRF and LIBS. The authors evidenced the advantage of LIBS in terms of spatial resolution (300 μm spot size for LIBS vs. 5 mm for XRF), which allows better discrimination between mixed minerals, besides the obvious fact that XRF is not able to detect lithium because of its low atomic number. In the study the authors used LIBS for acquiring elemental maps, which allowed the identification of different minerals in the samples (see Figure 7).

No attempt at lithium quantification was made in [59], but the following year the same group [60] analyzed a set of 124 geological samples from a mining site. The authors compared the precision of linear and non-linear methodologies for the quantification of lithium concentration in geological samples by LIBS.

They compared several univariate and multivariate linear and non-linear analytical approaches, using both a hand-held spectrometer and a laboratory setup for acquiring the LIBS spectra, finding that the laboratory LIBS setup outperformed the hand-held instrument and that non-linear methods gave more accurate results than linear methods. The authors also noticed that the use of non-linear methods reduced the gap in analytical performance between the laboratory setup and the hand-held instrumentation. The final result of their work was, however, that even the most performant non-linear approach, Support Vector Machine (SVM) and Artificial Neural Network (ANN), was not able to obtain errors lower than 30% in lithium concentration. This led to the conclusion that LIBS analysis showed clear limits for lithium quantitative analysis.

In spite of these results, the same authors [61] used non-linear methods for the quantification of Li concentration in 51 pegmatite samples from the Barroso mine in Portugal, obtaining very good correlation coefficients (0.97) against certified values.

Rifai et al. [62] used the LIBS ECORE analyzer realized by ELEMISSION Inc. (Montréal, QC, Canada) for determining the elemental map of Li in thirty crushed ores. An area of 25 × 150 mm^2^ was scanned, with a step size of 0.1 mm, in slightly more than 6 min per sample (see Figure 8).

Two methods were used for the quantitative analysis of lithium: the first involved the calculation of the spodumene percentage in the mineral maps and the determination of lithium from the empirical chemical formula, while the second consisted of a conventional univariate calibration approach using reference crushed samples.

The authors noted that both the methods yielded comparable results, but the conventional calibration curve approach was characterized by a lower bias and should thus be preferred. Moreover, they reported a relative standard deviation of the Li concentration measurements, less than 15% for Li concentrations higher than 0.6%, thus meeting the accuracy requirements for measurements at the cut-off lithium grade (about 0.6%).

Muller et al. [63,64] used ML methods for the classification of Li-bearing pegmatite and quantification of Li in drill core samples. For classification, the authors used a semi-supervised approach, exploiting the possibility of some technique, such as Linear Discriminant Analysis (LDA) and One-Class support vector machine (OC-SVM), of classifying objects in classes using a minimum number of reference mineral samples. The authors further improved the classification capabilities of their method using self-learning, i.e., the possibility for the algorithm to learn from the already classified samples for improving the discrimination between the Li-bearing minerals and the unknown matrix. They used this method for the analysis of drill cores from the Rapasaari area in Finland, obtaining excellent definition of the boundaries between Li-bearing minerals and other minerals. An optimum self-learning iteration of just one cycle was found to bring the best classification results.

João Manoel de Lima Júnior et al. [65] used a method called the Partial Matrix Matching Multi-Energy Calibration approach (PMM-MEC) with an internal standard for the determination of Li in spodumene samples from different locations in Minas Gerais, Brazil. The MEC procedure is essentially a variation in the traditional single-standard calibration method and, as such, it cannot compensate for variations in the matrix between reference and samples, not even partially. Nevertheless, in the analysis of two spodumene samples, the authors declared relative errors on Li concentration of the order of 2% with respect to the nominal concentration, with relative standard deviations between 0.6 wt% and 1 wt%.

Sweetapple and Tassios [66] used LIBS elemental mapping for characterizing petrographically pegmatite samples from the Mt. Cattlin deposit in Western Australia. The analysis of the samples allowed the authors to discriminate lithium (in the form of spodumene) against the silicate matrix of the rock. However, they noticed the difficulties in performing quantitative analysis due to the matrix difference between the lithium-doped borosilicate glass standards they used and the samples, as well as due to the self-absorption effect which is particularly important at Li concentrations larger than 2–6 wt%.

Umar et al. [67] quantitatively analyzed by LIBS the presence of Be, K and Na impurities in spodumene, using the standard addition method. They prepared the samples by fine grinding the spodumene samples and then mixed the powder with known amounts of beryllium, potassium and sodium oxides. The samples were then pressed to form solid pellets that were analyzed by LIBS to obtain three calibration curves, one for each element, whose linear extrapolation at zero signal intensity would correspond to the original concentration of the element in the sample. Typical relative errors in the concentration of the three impurity elements considered were of the order of 5%. The authors evidenced the simplicity of the method, which does not depend on the availability of certified matrix-matched samples, and the possibility of performing a direct analysis of the samples in powder form, without pelletization, for further speeding up the LIBS analysis.

Janovszky et al. [68] applied multivariate statistical analysis for classifying minerals according to lithium and beryllium content in granitoid rocks. They obtained high resolution elemental maps of these elements (see Figure 9).

They also quantified the Li and Be content in the rocks using a conventional univariate calibration curve based on matrix-matched standards. The authors reported the capability of the LIBS technique to detect Li at nanogram levels in the mineral grains.

Finally, Capela et al. [69] focused their research on the automatic classification of lithium-bearing mineral species. The authors used an unsupervised clustering method on the spectra obtained from LIBS micro-imaging of the samples to obtain elemental maps that are then transformed into mineralogical maps. The final result was a classification of the samples at the mineral scale; the authors evidenced some issues related to the presence of Li and Al traces in different minerals on spodumene, as well as incrustation of quartz and feldspar on spodumene, which might lead to misclassifications. At the same time, the limited resolution of the maps acquired (300 μm) might create classification problems in samples with fine granulometry. However, the observed issues could be resolved, according to the authors, by adjusting the experimental apparatus to obtain spatial resolutions on a scale of 10 μm.

### 3.2. Brines

About 40% of the global lithium production in 2024 originated from brine sources, as illustrated in Figure 2. These brines—typically extracted from salt flats, geothermal fluids and subsurface reservoirs—represent an increasingly important component of the lithium supply chain due to their high lithium content and comparatively lower environmental footprint. As demand for lithium continues to rise, understanding and monitoring these brine resources has become a priority for both industry and researchers.

Laser-Induced Breakdown Spectroscopy (LIBS) is particularly relevant in this context because the technique is known, at least in principle, to operate on solid, liquid and gaseous targets [70]. The ability of LIBS to directly interrogate different phases opens the door to a wide range of analytical applications, especially in environments where traditional sample preparation is difficult or where rapid, real-time measurements are required.

Because of this inherent flexibility, LIBS has been widely employed for determining lithium concentrations in liquid matrices, including natural and synthetic brines. Its capacity for rapid in situ chemical analysis allows for continuous monitoring of lithium extraction processes, quality control in brine processing streams and assessment of geochemical variability in fluid reservoirs.

However, the direct analysis of lithium in liquid solution can be problematic [71], suggesting in some cases the conversion of the liquid into a solid by freezing the solution [72] or making the liquid dry on a surface, as in the Surface-Enhanced LIBS (SENLIBS) technique [73] or when using micro-extraction procedures [74].

Berlo et al. [75] adopted the strategy of freezing the samples at −30 °C for analyzing volcanic brines with a tandem LIBS-ICP-MS instrument. The authors noted that, although freezing the liquid sample improves the coupling with the laser energy, leading to more intense and reproducible LIBS spectra, in the case of brines, freezing is made difficult by the high concentration of salts dissolved in the liquid sample. Moreover, changes in the sample’s matrix may affect the LIBS signal due to the possible variations in the hardness of the frozen liquid. After careful optimization of the experimental conditions and normalization of the LIBS signal using Al as an internal standard, the authors evaluated a 10% uncertainty in the measure of Li concentration in brine.

Alternatively, the liquid sample can be transformed into an aerosol and sprayed in a spray chamber, as Xing et al. [76] did for the analysis of lithium in brine. The authors used argon as a carrier gas, although they also suggested the possibility of using nitrogen or air. For the quantitative analysis they chose a deep learning approach [49,50], testing two convolutional neural networks (CNN) built considering as input the emission lines of the major matrix elements (K, Na, Ca and Mg) plus one Li emission line (670.8 nm) or using only the two Li lines at 610.35 and 670.8 nm, respectively. The training of the CCN was performed using simulated brine solutions, while five real brine samples, collected from the Qinghai and Xinjiang Chinese provinces, were used for external validation of the model.

The authors found that the CNN built including the brine matrix elements was outperformed by the one having as input the two lithium lines only, which reached an average absolute error around 0.3 mg L^−1^ and an average relative error of 3.5%. The Limit of Detection (LOD) for Li detection in brine was determined to be 0.7 mg L^−1^, a value that is lower than the mining grade brine for lithium.

Erbetta et al. [77] used a Venturi system for nebulizing Li-rich brine and analyzed it with a hand-held LIBS instrument. The reproducibility of the signal obtained was much better than the one obtained by focusing the laser beam on the liquid brine surface, despite the lower intensity of the signal. The authors analyzed 13 brine samples from the Atacama Desert in Chile, using a univariate calibration curve method. The Li concentration in the standards used for calibration ranged from 0.4 to 1.5 wt% in a 15% NaCl matrix. They reported an accuracy on the Li determination ranging from 2 to 20%, with a relative RSD lower than 10% in normal operation conditions.

Kardamaki et al. [78] analyzed brine by focusing the laser at the surface of the liquid flow. The authors calibrated their measurements using artificial saline solutions in the range between 10 and 1300 mg/L, obtaining a good linearity for the Li emission line at 670.8 nm with an *R*^2^ value of 99.98% of the calibration curve. They reported average relative uncertainties in the analysis of Li in high saline fluids that were lower than 2%, with the precision of the analysis ranging from 0.1% to 7%.

Molina et al. [79] analyzed brines collected from different *salares* located in the Puna region of Argentina, in the ‘Lithium triangle’. For the quantitative analysis, the liquid brine was converted to solid in the form of pressed pellets from pure calcium hydroxide [80]. The authors used a univariate calibration approach; the calibration curve was built from 11 standard solutions prepared by diluting a stock solution of lithium carbonate with appropriate volumes of doubly distilled water. For the quantification of Li concentration, they used the Li I emission lines at 670.8 nm, normalizing the signal by the intensity of the Ca I line at 671.8 nm. The calibration curve was fitted with a saturating exponential function; according to the authors, the curve maintained an almost linear behavior at Li concentrations < 400 μg/g. After that concentration, the self-absorption effect could not be neglected anymore. The average error in the quantification of Li in brine was evaluated by the authors to be around 25%, with the quantification error increasing with the increase in Li concentration due to the self-absorption effect causing the saturation of the calibration curve. The estimated LOD was 0.2 μg/g.

### 3.3. Batteries

Several recent papers have focused on the use of LIBS for the analysis of lithium in the context of resource recovery and recycling [81]. LIBS can be used for rapidly identifying lithium in complex, heterogeneous waste such as spent batteries, industrial residues and metallurgical by-products. The LIBS capability of providing fast, multi-element detection with minimal sample preparation enables efficient screening of materials that contain lithium-bearing phases. This is especially valuable in recycling environments, where materials often vary significantly in composition and require real-time or near-real-time characterization.

Galli et al. [48] used an artificial neural network (ANN) for quantifying lithium concentration in black mass using hand-held LIBS instrumentation. Black mass is a carbon-based material resulting from the recovery process of spent lithium-ion batteries.

The notable aspect of this paper is the evidence that a single ANN can be built for quantifying lithium concentration in different matrixes (graphite + LiCoO_2_, LCO and graphite  +  LiMnO_4_, LMO) after a thoughtful selection of the input spectral features. The unified model was characterized by a Root Mean Square Error (RMSE) of 0.33% in the range of Li concentrations between 0.2 wt% and 4.5 wt% [64]. Pourmohammad et al. [82] also analyzed black mass from spent Li-ion scooter batteries with several analytical techniques, including LIBS, obtaining qualitative information about the distribution of Li and other elements in black mass graphite and metallic particles.

The same advantages that make LIBS interesting for the analysis of lithium recovery and recycling have also been exploited for the analysis of Li-rich materials used in industry.

For example, Chen et al. [83,84] used femtosecond LIBS, among several analytical techniques, for studying the relationships between the surface chemistry and interfacial behavior in lithium cells. The authors reconstructed in-depth Li maps on dense LLZO (Al-substituted Li_7_La_3_Zr_2_O_12_) pellets processed in controlled atmospheres. The authors noticed that in an LLZO pellet aged in air for two months, a Li-rich layer of about 3.5 μm was present at the surface, which the authors attributed to the formation of Li_2_CO_3_ from the interaction of Li in the sample with ambient CO_2_.

Pinson et al. [85] studied the degradation of lithium hydride (LiH) and anhydrous lithium hydroxide (LiOH) when exposed to humidity and CO_2_. The samples were placed in an environmental chamber where they were analyzed in a combined LIBS and Raman experiment, using a pulsed laser and an Echelle spectrometer. The LIBS and Raman data were then fused together and analyzed with an ML algorithm; this allowed the authors to better understand the weatherization processes of Li compounds.

Hou et al. [86] used a femtosecond laser for reconstructing three-dimensional images of Li-ion solid-state electrolyte by LIBS, while Zheng et al. [87] used LIBS for understanding the influence of 3D anode architecture on lithium distribution during charging and discharging at elevated C-rates. In a post-mortem study the authors obtained the 3D distribution in a graphite electrode, scanning the surface with 100 μm steps and an ablation rate of about ~6.4 µm per layer, on average. The 3D map, reproduced in Figure 10, evidences a homogeneous distribution of lithium, except in the first two layers (from top to bottom) where a higher concentration of lithium is observed. In these layers a radial lithium diffusion is also noticeable, with lithium accumulating along the contour of the structure.

Imashuku et al. [88] used LIBS for quantitative Li mapping of lithium-ion battery cathodes containing LiCoO_2_ as an active material. They performed the measurements in a reduced Ar atmosphere (1 kPa) and used the 610.4 nm Li emission line for building a calibration curve at different ratios of lithium to cobalt. The authors noticed that the measure of the Li/Co concentration in the 30- and 50-cycle cathodes was rather imprecise, possibly because of the porosity of the cathode; however, the simplicity of LIBS analysis allowed the authors to appreciate the difference between the relatively homogeneous cathode cycled 30 times and an inhomogeneous distribution with parts of the cathode along the edges and near the center with a Li/Co ratio > 1. This result was evidence that a lithium compound other than LiCoO_2_ precipitated on the cathode cycled 50 times. The interpretation of the authors was that, following cathode overcharge, LiCoO_2_ dissolved into the LiPF_6_ electrolyte, causing the precipitation of LiF and PF_5_, the decomposition products of the electrolyte. Therefore, despite the poor precision of the quantitative results, LIBS was able to evidence the effects of battery overcharging.

The same authors [89] also mapped Li on the graphite anode in three dimensions with the same experimental setup used in [88]. The in-depth analysis was performed by repeating the laser irradiation several times at the same point. Imashuku et al. observed a homogeneous or inhomogeneous Li distribution after a charge–discharge cycle, depending on the charge–discharge curve of the cells and the lithium-ion transfer mechanism in the graphite anode.

Imashuku et al. [90] proposed the use of LIBS for determining the state of charge (SOC) distribution of Li-ion battery cathodes, on the basis of the results obtained in [88] on battery cathodes. The SOC of the charged cathode obtained using the LIBS technique was in agreement with the average SOC estimated from the charge–discharge curves.

Imashuku [91] also applied LIBS for studying the lithium penetration into refractories used for recovering cobalt from spent Li-ion batteries. The author compared alumina (Al_2_O_3_) and magnesia (MgO) crucibles for recovering cobalt from a model Li-ion battery. The crucibles were heated at 1500 °C in Ar atmosphere and the Li distribution was obtained by LIBS at a depth of 0.5–1 μm from the surface. They observed that Li penetrates both alumina and magnesia crucibles. However, repeated use of the Al_2_O_3_ did not show signs of damage because of the formation of compounds such as LiCoO_2_, LiAlO_2_, CaAl_4_O_7_ and CaAl_2_O_4_ which acted as a protection against further penetration of Li. On the other hand, cracks formed in the MgO crucible after the first use because of the diffusion of Li in MgO. The author concluded that an Al_2_O_3_-based refractory should be preferred to an MgO one for the recovery of metals from spent Li-ion batteries.

### 3.4. Other Li Rich Materials

A significant portion of lithium consumption is destined for the glass and ceramics industry. LIBS has gained increasing attention for the analysis of lithium-containing materials where lithium plays a key role in modifying the physical and chemical properties of final products. Lithium compounds are widely incorporated into glass formulations to improve thermal shock resistance, reduce viscosity and enhance melting behavior. In ceramics, lithium contributes to the development of specialized phases and influences properties such as translucency, dielectric performance and mechanical strength. Accurate, rapid determination of lithium content in raw materials is therefore essential for ensuring product quality and consistent manufacturing outcomes.

LIBS is particularly well suited for analyzing these lithium-bearing materials because of its strong sensitivity to light elements, which are often challenging to quantify using conventional analytical techniques

Recent studies have focused on evaluating the potential for lithium recovery from industrial materials such as lithium aluminosilicate glass and industrial enamels, which contain approximately 10% and 1% lithium, respectively. However, the feasibility of such extraction depends on the ability to rapidly and accurately determine lithium content in a manner applicable to industrial-scale and in situ contexts, thereby avoiding the time-, labor- and material-intensive procedures required by traditional liquid matrix analyses. In this perspective, an ANN-based approach was used by Raneri et al. [92] for the analysis of lithium in non-compliant materials from the enamel industry. In these materials, lithium generally acts as a glass modifier, usually incorporated as lithium oxide (Li_2_O) in amounts of about 1–3 wt.%. Such materials consist of micron-sized powders that are electrostatically applied to metal surfaces (like oven bodies or other household and industrial components) and subsequently fired to produce a smooth, resilient finish. However, during manufacturing or firing, some enamel batches can develop technical flaws that make them unsuitable for their intended use, categorizing them as non-conforming products. Recovering lithium from these rejected materials presents both environmental and economic advantages, supporting circular economy principles and promoting zero-waste production. Using ML algorithms, the authors succeeded in building a single model capable of predicting Li concentration in enamel samples characterized by very different matrixes. A RMSE lower than 1 wt% was obtained in a range of Li concentrations ranging from 0.5 to 5 wt%.

The extraction of lithium from lithium-rich ceramic and glass materials is a relatively new area of research, and to the authors’ knowledge, this study represents the first application of LIBS for the in situ quantification of lithium-rich industrial secondary raw materials from a recycling perspective.

### 3.5. Biological Systems

Several groups have analyzed by LIBS the presence of lithium in human or animal tissues and its effect on plants. Lithium has a dual role in plant development, because while low concentrations can promote growth, elevated levels become harmful by triggering plant oxidative stress. Lithium also has important effects on biological systems. From a medical point of view, lithium is widely used to treat bipolar disorder; on the other hand, excessive lithium intake might cause kidney and thyroid dysfunctions.

#### 3.5.1. Animals

Irfan Ahmed et al. [93] proposed LIBS as a rapid detection method for detecting lithium in biological tissues. They also used LIBS for studying the transfer of lithium from mammary glands to milk in breast-feeding rats administered with lithium treatment [94,95]. The authors detected lithium at the level of 1 μg/g in the mammary glands of treated rats, while it was below the limits of detection in controls. In another study on pups breast-fed by lithium-treated rats [96], the same authors observed symptoms of hypothyroidism that persisted after lithium was cleared from the blood. They attributed the disease to lithium interference with the sodium-iodine mechanism, preventing thyroid iodine uptake. The results of this study stressed, according to the authors, the importance of integrating iodine supplements into the diet of breast-feeding women under lithium treatment to counterbalance the negative effect of lithium on the infant’s organism.

Lithium chloride has also been recently proposed as a tumor-labeling agent by Oğuz Gürcüoğlu et al. [97]. The authors demonstrated the possibility of identifying and ablating the LiCl-marked tumor cells using LIBS, suggesting the use of the technique for developing a dual-purpose platform for precise cancer detection and resection through laser ablation.

#### 3.5.2. Plants

In a study on lithium intake in plants, Singh et al. [98] used LIBS for obtaining elemental maps of *Podocarpus macrophylla* (yew plum pine) leaves, analyzing the distribution of lithium dioxide in the leaves. Their study found that lithium dioxide, absorbed by the roots from the soil and transported to the leaves by the xylem, is released back through the leaf veins. Through this mechanism, the plant can regulate the flux of lithium into its leaves.

### 3.6. Other Applications

#### 3.6.1. Isotopic Analysis

One of the earliest studies on lithium using portable instrumentation was performed by Cremers et al. [99] which studied the possibility of determining the ^6^Li/^7^Li isotope ratio in the framework of nuclear forensic applications. The authors used a compact LIBS instrument coupled with a high-resolution Echelle spectrometer; however, due to the large width of the main Li emission lines and the small difference between peak emission of the two isotopes (15.8 ± 0.3 pm), they were only able to determine the isotope ratio ofhighly enriched samples, with ^6^Li/^7^Li ~1 (the natural ^6^Li/^7^Li isotope ratio is around 0.08).

Wood et al. [100] improved the sensitivity of the technique using a spectrometer with 70 pm full width at half maximum (FWHM) instrumental broadening and chemometric analysis. Although the separation between the ^6^Li and ^7^Li peaks was much smaller than the instrumental broadening, the authors were able to obtain a consistent evaluation of the isotope ratio in the range between 3% and 85.5% with good linearity (R^2^ > 0.90) and precision (RMSE < 0.09). They found that a non-linear method based on Artificial Neural Networks would give better results, especially at high isotope ratios, than linear methods such as Principal Component Analysis (PCA) and Partial Least Square (PLS) analysis.

Hull et al. [101] combined Laser Ablation–Tunable Diode Laser Absorption Spectroscopy (LA-TDLAS) and LIBS for lithium isotope analysis and plasma diagnostics. The authors exploited the high spectral resolution of TDLAS for determining the Li isotopic ratio, while LIBS was used for plasma diagnostics (electron number density and temperature).

An interesting method for measuring the ^6^Li/^7^Li isotopic ratio has been proposed by Touchet et al. [102]. The method, called Laser-induced Breakdown Self-reversal Isotopic Spectrometry (LIBRIS) exploits the self-reversal phenomenon of the 670.8 nm Li line, correlating the position of the self-reversal ‘dip’ with the ^6^Li/^7^Li isotopic ratio. Gallot-Duval et al. [103] improved the precision of ^6^Li determination, obtaining measurement errors lower than 5–10% using laser spots of 250 μm. The uncertainty on ^6^Li remained acceptable (25–30%) in high-resolution applications, where laser spots of 7 μm were used.

Moran et al. [104] further improved the potential of the LIBRIS technique using supervised ML for determining the link between the measured shift in the self-reversed ‘dip’ and the isotope ratio. Staking together the predictions of multiple ML algorithms, the authors demonstrated an RMSE of results lower than 6% in the interval of ^6^Li numerical concentrations between 3 and 95%, with an estimated LOD lower than 20%.

One of the strong requirements of the LIBRIS approach is the need for high spectral resolution spectrometers, capable of resolving the minimal isotopic shifts expected in ^6^Li-^7^Li systems. In a recent paper, Tran et al. [105] were able to obtain ^6^Li isotope analysis with Li abundances ranging from 1% to 95% using a combination of Laser-produced Vapor (LPV) and LIBS analysis. They used a 532 nm laser, focused on the liquid sample 3 mm under the surface, for vaporizing a ^6^Li/^7^Li solution. A second laser beam at 1064 nm was used for the LIBS analysis of the vapor, 7 mm above the liquid surface. The LIBS signal was acquired using a spectrometer with an optical resolution of 25 pm, larger than the maximum isotope shift in the 670.8 nm Li line. After optimization of the experimental conditions to avoid the self-reversal effects of the Li spectral lines, the authors were able to measure the red shift in the 670.8 nm line peak position with the increase of ^6^Li concentration, obtaining a very good linear correlation (R^2^ = 0.998) between spectral shift and ^6^Li concentration in the whole range between 1 and 95%. The authors measured the isotopic shift between ^6^Li and ^7^Li as 15.7 pm, very close to the theoretical value of 15.8 pm.

#### 3.6.2. Nuclear Applications

Lee et al. [106,107] used double-pulse LIBS (DP-LIBS) for detecting lithium in aqueous solution. In DP-LIBS, two laser pulses are sent sequentially on the target. In the case of ref. [106,107], the two laser pulses were collinear and focused on the surface of a liquid jet. In collinear DP-LIBS, the hot gas behind the shock wave produced by the first laser pulse reduces the plasma shielding and creates an optimum environment for the propagation of the second laser pulse [108]. The strong enhancement of the LIBS signal obtained using a DP-LIBS configuration allowed the authors to obtain an LOD for Li of 0.8 ppb. This level of sensitivity was considered sufficient for proposing LIBS as an online elemental analyzer of lithium in pressurized water nuclear reactors.

Monitoring Li concentration in nuclear reactors is important because of the corrosive properties of Li solutions on the surface of the metal structure materials used in nuclear reactors. Li et al. [109,110,111,112,113] used LIBS for depth-profiling of corroded stainless steel, obtaining the profiles of steel elements and corrosion medium Li from the relative intensities of the corresponding emission lines. They observed a corrosion layer of about 2 microns at the surface, characterized by an enrichment in Fe, Ni and Mn, and a corresponding depletion of Cr. Lithium was observed up to a depth of about 4 microns under the steel surface.

The same authors [114] studied the effect of Er_2_O_3_ ceramic coating in preventing Li corrosion on steel surfaces. They used two different strategies for depositing the coating, the first by depositing one erbium layer on the steel sample and oxidizing it successively, and the second by direct reactive sputtering with Er_2_O_3_ deposition. The authors realized that the first approach protected the steel surface more effectively, preventing Li from arriving at the metal surface. However, it was also prone to microcracks and defects, which might allow Li penetration. Conversely, the second type of treatment, although effective in protecting the surface, tended to break down and peel off from the sample because of the imperfect bond between the oxide film and the surface of the metal.

Feng et al. [115] studied the Li corrosion on a steel sample treated with Er_2_O_3_ coating deposited by sol–gel method. The corrosion resistance of the coating was measured, determining the in depth Er/Li concentration ratio as a measure of the penetration depth of Li in the coating. CF-LIBS was used for determining the ratio independently on the matrix changes in the sample. The authors found that the coatings prepared at temperatures of 650 and 800 °C effectively protected the steel surface; however, the coating prepared at 950 °C showed worse performance in lithium corrosion resistance because of the increased porosity of the coating.

#### 3.6.3. Planetary Exploration

One of the key developments in LIBS research has been the realization of instrumentation capable of working outside our planet. At present time there are three LIBS instruments on Mars’s surface, two of them are mounted on the NASA rovers Curiosity [116,117] and Perseverance [118,119] and one is on the Chinese rover Zhurong [120]. Like many other elements, lithium has been detected and quantitatively analyzed by LIBS on the surface of Mars. Ollila et al. [121] built a univariate calibration curve for analyzing the LIBS spectra obtained by the ChemCam instrument of the Curiosity rover based on the intensity of the Li line at 670.8 nm. Using 32 standards, they obtained an RMSE of 36 ppm. The authors also tested a multivariate linear model (Partial Least Square) using selected parts of the LIBS spectrum, obtaining results similar to the ones of univariate calibration. An improved univariate calibration curve was obtained by Payré et al. [122], which reached RMSE values of 5 ppm, using 88 standards.

The accuracy of lithium quantification in Mars-equivalent conditions was studied by Ytsma and Dyar [123], who analyzed 402 rock standards by LIBS and used univariate calibration and multivariate analysis for quantifying several light elements. For lithium univariate analysis, the authors used the intensity of the 670.8 nm Li line, while multivariate analysis was performed using two linear methods, PLS and Least absolute Selection and Shrinkage Operator (LASSO). The authors found that both univariate and multivariate approaches were characterized by similar RMSE (around 25 ppm) but noticed that the multivariate approach was preferable because of the better linearity of the calibration curves. The MarSCoDe LIBS instrument mounted on the Zhurong rover was capable of quantifying Li concentrations ranging from 6 ppm to 18 ppm using a univariate calibration model built in the ground laboratory and then validated using 12 onboard calibration samples [124].

## 4. New Approaches to LIBS Analysis of Lithium

Research on LIBS over the last decade has shown that one of the most effective routes to improving the analytical capabilities of the technique lies in the development and application of advanced chemometric tools based on machine learning algorithms. The ability of LIBS to generate large amounts of data in a very short time, initially regarded as a challenge, has increasingly come to be seen as one of its greatest assets. Indeed, the machine learning methods developed in recent years benefit greatly from the redundancy of information obtainable during analysis, both for achieving precise classifications and for performing accurate quantitative measurements.

It can be said that many of the recent advances in the LIBS analysis of lithium in complex matrices, whether for the characterization of recycling materials or for its determination in geological samples, would not have been possible without the use of advanced algorithms that apply machine learning principles through multivariate and non-linear analysis of the relationships between the LIBS spectral data and the physical and chemical properties of the system.

Table 4 summarizes the main approaches used for the quantitative analysis of lithium compounds, along with the relevant analytical figures of merit. When multiple methods are used within the same study, the best analytical results are reported.

Due to the lack of standardization in how LIBS results are presented, the analytical indicators provided may vary depending on the study and its specific application. In addition to the *LOD*, which is usually calculated from Equation (1), and the *RMSE*, defined in Equation (2), other important figures of merit include the slope *b* of the predicted-versus-reference calibration curve and the correlation coefficient $R^{2}$, defined as follows:(3)$$R^{2}=1-\frac{\sum_{i=1}^{n}{\left(p_{i}-r_{i}\right)}^{2}}{\sum_{i=1}^{n}{\left(p_{i}-\bar{p}\right)}^{2}}$$
where n is the number of samples, $p_{i}$ are the predictions obtained from the experiment, $r_{i}$ are the corresponding reference values and $\bar{p}$ is the mean of the experimental predictions.

Other parameters that are often reported are the Mean Absolute Error (*MAE*):(4)$$MAE=\frac{\sum_{i=1}^{n}\left|\left(p_{i}-r_{i}\right)\right|}{n}$$ the Mean Absolute Percentage Error (*MAPE*):(5)$$MAPE=\frac{100}{n}\sum_{i=1}^{n}\left|\frac{\left(p_{i}-r_{i}\right)}{p_{i}}\right|$$ and the Mean Squared Error (*MSE*):(6)$$MSE=RMSE^{2}$$

It can be observed that, in some cases, a conventional approach based on matrix-matched standards for constructing univariate calibration curves is still capable of delivering excellent quantitative results. However, in many challenging situations, especially when the sample matrix is expected to vary substantially from one specimen to another, the use of powerful multivariate machine learning algorithms becomes the preferred and, in some cases, the only viable solution.

## 5. Conclusions

Over the past 15 years, numerous studies have explored Laser-Induced Breakdown Spectroscopy as a promising technique for lithium detection. Conventional laboratory methods such as AAS, ICP-OES and ICP-MS offer excellent detection limits but are hindered by complex sample preparation and lack of suitability for rapid or in situ analyses. These constraints have driven interest in LIBS as a fast, portable and reagent-free alternative capable of providing real-time, multi-elemental information across the lithium value chain which can also be competitive against other emerging spectroscopic methods (see Table 5).

Among these techniques, Raman spectroscopy is probably the most mature for lithium analysis, given its capability to characterize Li phases non-destructively and with excellent spatial resolution. However, the Raman signal is weak and can be easily overwhelmed by fluorescence. Moreover, the technique is rather slow, and the Raman surface mapping of materials, although very rich in information, takes substantially longer than LIBS elemental mapping. Finally, Raman spectroscopy cannot provide quantitative information and is limited to surface analysis [125].

LIBS has proven its versatility in geological exploration, brine monitoring, industrial process control, recycling and even biological and extraterrestrial applications. Advances in data analytics, particularly through machine learning (ML) and deep learning (DL), have substantially enhanced its quantitative accuracy by compensating for matrix effects, improving calibration robustness and facilitating interpretation of complex spectra.

Nonetheless, the widespread adoption of LIBS for lithium determination remains constrained by several factors, which are schematically reported in Table 6.

The absence of standardized calibration protocols and certified reference materials (CRMs) for complex matrices such as ores, brines and battery waste hampers reproducibility and interlaboratory comparability. Strong matrix dependencies linked to mineralogy, surface roughness, porosity and spectral interferences further complicate quantification. Although ML and DL methods mitigate these effects, their performance depends on extensive, representative datasets and raises concerns about model transferability across instruments and conditions.

Additional challenges include limited sensitivity for highly diluted or compositionally complex samples and difficulties in determining lithium isotope ratios (^6^Li/^7^Li), which demand high spectral resolution and stable plasma conditions. These issues become even more critical in extraterrestrial environments, where calibration opportunities are minimal. Overcoming these barriers requires standardized methodologies, transferable ML/DL frameworks and integration with complementary analytical techniques to establish LIBS as a reliable tool for lithium monitoring and decision-making.

Within the framework of the European Critical Raw Materials Act (CRM Act, 2023), LIBS aligns with the goals of sustainable and transparent lithium supply chains. Its portability, reagent-free operation and rapid multi-elemental capability make it ideal for in situ and in-line verification across extraction, processing and recycling stages, ensuring analytical traceability and environmental accountability.

Future developments will likely transform LIBS from a diagnostic technique into a smart analytical platform through the convergence of intelligent algorithms, sensor fusion and automated process control. Such integration will support sustainable lithium management on Earth and precise, adaptive monitoring in space exploration.

To ensure industrial and regulatory compatibility, harmonized analytical protocols and interlaboratory calibration procedures are essential. Currently, no ISO or ASTM standard governs LIBS-based lithium quantification across the value chain. Establishing matrix-specific Standard Operating Procedures (SOPs), certified reference materials and dedicated round-robin validation studies is crucial for defining accuracy, reproducibility and comparability with ICP-based reference methods. The development of standardized performance metrics—covering detection limits, accuracy, spectral quality and plasma diagnostics—together with harmonized data-reporting templates and open spectral formats, will ensure interoperability and transparency. These measures will enable LIBS to fully meet the traceability and reliability objectives set by the EU Critical Raw Materials Act.

## Figures and Tables

**Figure 1 sensors-25-07689-f001:**
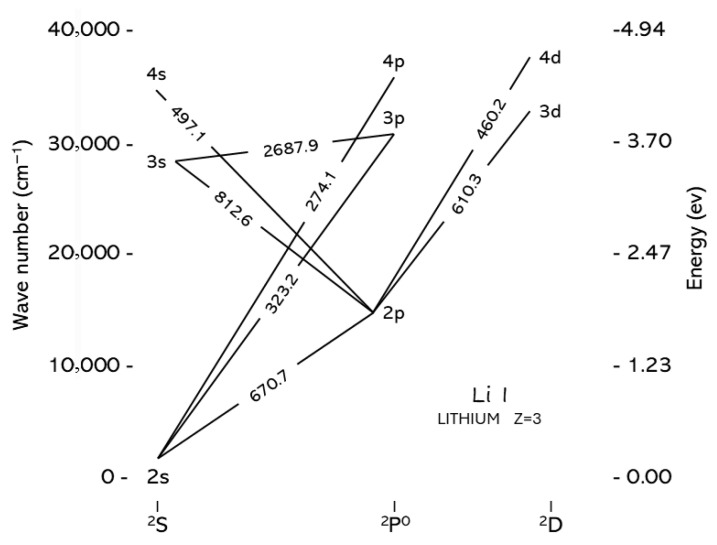
Grotrian diagram of Li I.

**Figure 2 sensors-25-07689-f002:**
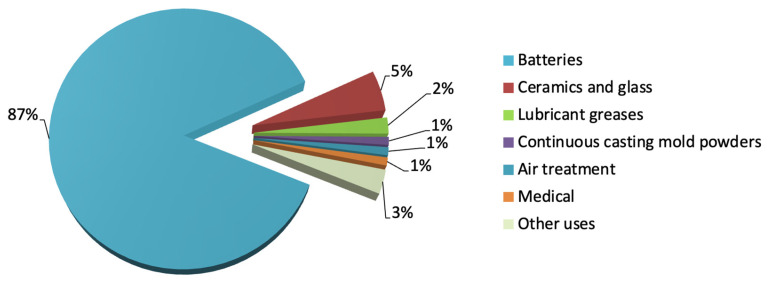
Industrial uses of lithium (2024 data) [7].

**Figure 3 sensors-25-07689-f003:**
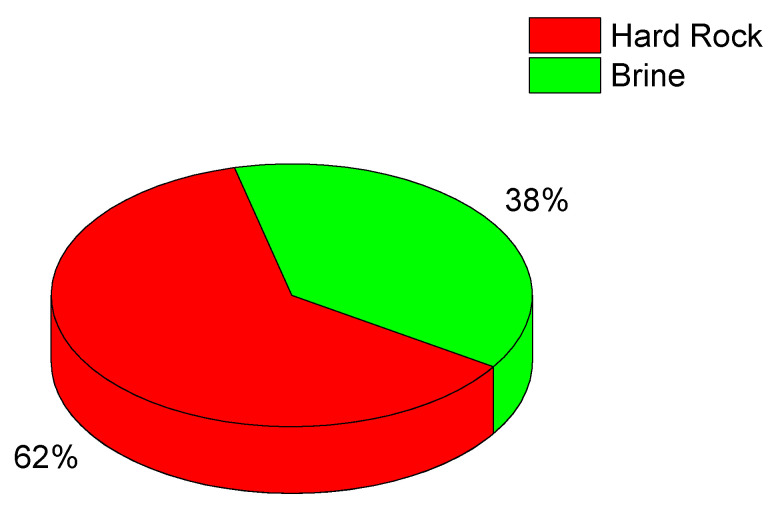
Sources of lithium (2024 data) [6].

**Figure 4 sensors-25-07689-f004:**
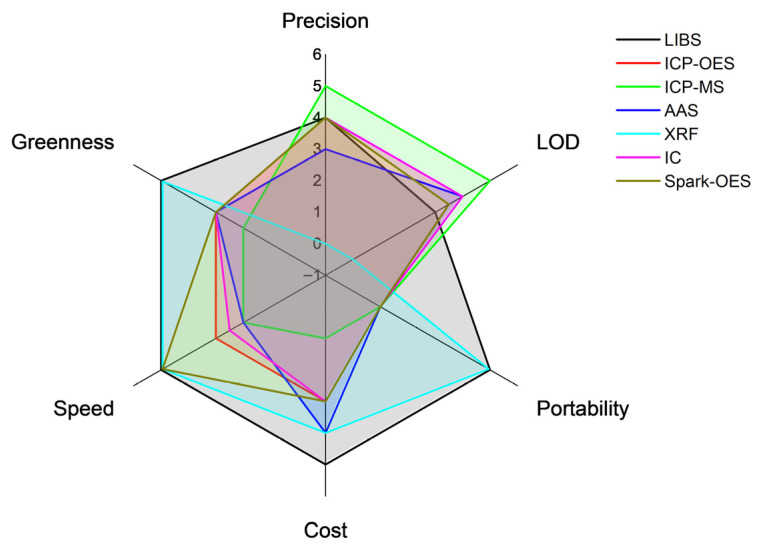
Graphical representation of the analytical performances of the main techniques that can be used for the analysis of lithium compounds.

**Figure 5 sensors-25-07689-f005:**
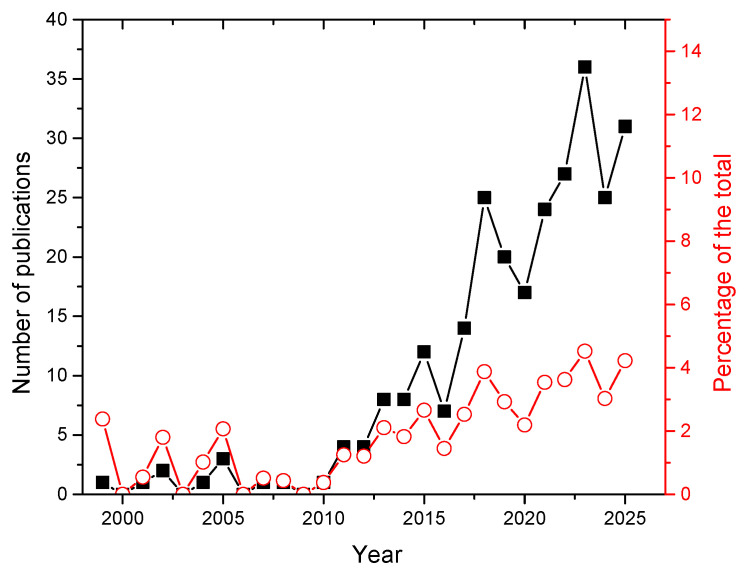
Number of papers on LIBS applications on lithium published per year (black points, left scale) and percentage of the total publications on LIBS (red point, right scale). Source: Scopus™.

**Figure 6 sensors-25-07689-f006:**
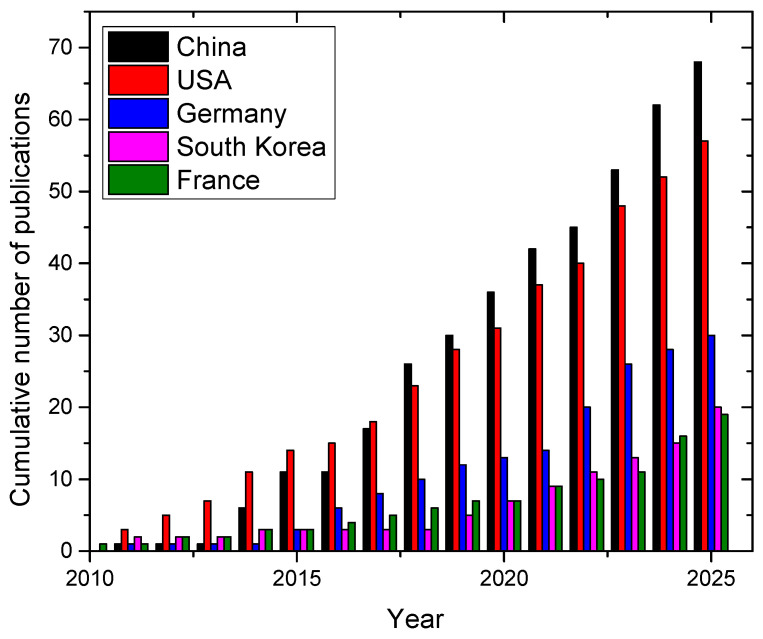
Cumulative number of papers on LIBS applications on lithium published per year by country (data from 2010 to 2025). Source: Scopus™.

**Figure 7 sensors-25-07689-f007:**
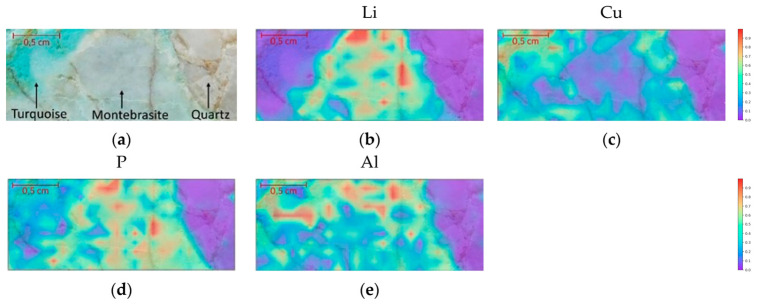
Elemental maps obtained by LIBS on sample (**a**). (**b**) Li (812.62 nm); (**c**) Cu (327.37 nm); (**d**) P (213.51 nm) and (**e**) Al (369.09 nm). The scale represents normalized intensity counts. Reproduced from [59] under open access Creative Common CC BY license.

**Figure 8 sensors-25-07689-f008:**
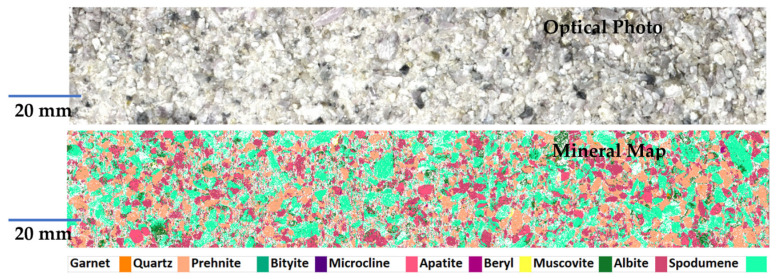
Mineral maps obtained using the LIBS ECORE analyzer (Elemission, Saint-Laurent (QC, Canada). The green zones correspond to Li-rich Spodumene. Reproduced from [62] under open access Creative Common CC BY license.

**Figure 9 sensors-25-07689-f009:**
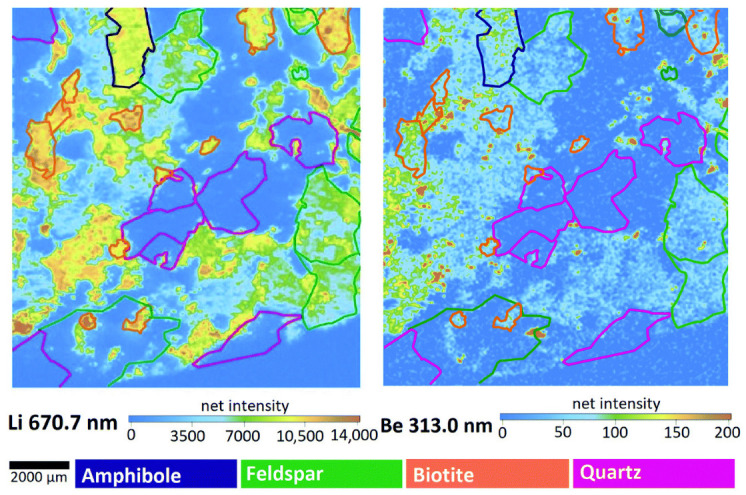
Elemental maps of Li and Be in Mórágy Granite rock. The color of the outlines defines the different minerals. Reproduced from [68] under open access Creative Common CC BY license.

**Figure 10 sensors-25-07689-f010:**
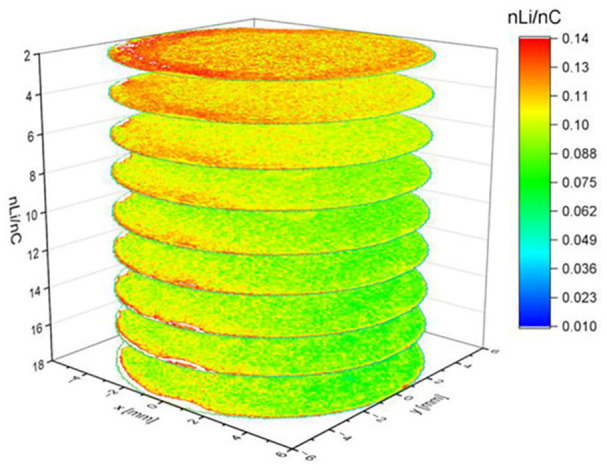
A 3D reconstruction of Li concentration within the graphite electrode. Figure reproduced from ref. [87] under open access Creative Common CC BY license.

**Table 2 sensors-25-07689-t002:** Lithium production and total reserves (2024 data).

	Production per Year (Ktons)	Percentage	Estimated Lithium Reserves (Mtons)
Australia	88	37%	7
Chile	49	21%	9.3
China	41	17%	3.0
Zimbabwe	22	9%	0.48
Argentina	18	8%	4
Brazil	10	4%	0.39
Canada	4.3	2%	1.2
Namibia	2.7	1%	0.014
Portugal	0.38	0.15%	0.06
USA	-	-	1.8

**Table 3 sensors-25-07689-t003:** Analytical performances of the main techniques that can be used for the analysis of lithium compounds.

Technique	Main Application Matrices	Typical *LOD*	Precision (*RSD/RMSE*)	Average Analysis Time	Operational Cost	Portability
LIBS	Rocks/Minerals/Brines/Black mass	0.2–0.7 mg/L (liquids); ~0.6 wt% (solids)	2–15% (matrix-dependent)	5–10 min	Low	High (hand-held, portable)
ICP-OES	Liquids, brines, digested solids	0.01 mg/L	<5%	30–60 min	Medium	Low
ICP-MS	Solutions, brines, biological samples	<0.001 mg/L	<3%	1–2 h (including prep.)	High	Low
AAS	Liquids, digested black mass	0.01–0.1 mg/L	5–10%	1–2 h	Low	Low
IC	Saline and biological solutions	~0.01 mg/L	<5%	45–90 min	Medium	Low
Spark-OES	Metallic samples, liquids	~0.1 mg/L	<5%	5–10 min	Medium	Low
XRF	Rocks/Minerals	Not applicable (Li not detectable)	-	5–10 min	Medium	High (hand-held, portable)

**Table 4 sensors-25-07689-t004:** Summary of the analytical methods and factors of merit in LIBS analysis of lithium.

Paper	Analysis	Analytical Tool	Spectral Features	Analytical Indicators
Korbel et al. [23]	Geological samples	Multivariate linear regression	Multivariate	*R*^2^ = 0.94 *RMSE* = 0.15 wt% *MAPE* = 11.75%
Fabre et al. [40]	Geological samples	Univariate calibration curve	610.36 nm and 670.79 nm	*LOD* = 0.1 ppm *R*^2^ > 0.9 *RMSE* = 0.2 wt%
Galli et al. [48]	Black Mass	Machine learning (ANN)	Multivariate	*R*^2^ = 0.94 *RMSE* = 0.33 wt%
Fabre et al. [52]	Geological samples	Univariate calibration curve	670.79 nm	*LOD* = 5 ppm
Mezoued et al. [54]	Geological samples	Univariate calibration curve	Ratio of Li signal to the matrix elements	*RMSE* = 0.062 wt% *MAE* = 0.081 wt%
Horňáčková et al. [55]	Geological samples	CF-LIBS	Multi-elemental analysis	Not reported
Ribeiro et al. [59]	Geological samples	Univariate calibration curve	812.62 nm	*LOD* = 65 ppm
Ferreira et al. [60]	Geological samples	Machine learning (ANN)	Multivariate analysis	*MAPE* = 34% *MAE* = 1339 ppm
Guimarães et al. [61]	Geological samples	Machine learning	Multivariate analysis	*R* = 0.97
Rifaï et al. [62]	Geological samples	Univariate calibration curve	610.36 nm	*R*^2^ = 0.982 Slope = 0.998
de Lima Júnior et al. [65]	Geological samples	PPM-MEC with Internal Standard	323.27 nm, 610.36 nm, 670.79 nm with Na I 589.6 nm as internal standard	*MAPE* = 4% Slope = 0.96
Xing et al. [76]	Brine	Deep Learning (CNN)	610.35 nm, 670.79 nm	*LOD* = 0.7 mg L^−1^
Erbetta et al. [77]	Brine	Univariate calibration curve	670.79 nm	*LOD* = 13 mg kg^−1^ *MAPE* < 5%
Kardamaki et al. [78]	Brine	Univariate calibration curve	670.79 nm	*R*^2^ = 0.998 *MAPE* < 2%
Molina et al. [79]	Brine	Univariate calibration curve	670.79 nm, normalization with Ca I line at 671.8 nm	*LOD* = 0.2 μg/g
Imashuku et al. [88]	Batteries	Univariate calibration curve (Li/Co)	610.4 nm	*RSD* < 7%
Raneri et al. [92]	Enamel	Machine Learning (ANN)	Multivariate	*MAPE* = 14% *RMSE* = 0.97 wt%
Ahmed et al. [93]	Biological tissues	Univariate calibration curve	670.79 nm	*LOD* = 0.1 ppm
Ahmed et al. [94,95]	Biological tissues	Univariate calibration curve	670.79 nm	*LOD* = 0.1 ppm
Ahmed et al. [96]	Biological tissues	Univariate calibration curve	670.79 nm	*LOD* = 0.007 µmol/L
Cremers et al. [99]	Liquid samples	Line fitting	670.79 nm	*LOD* < 50%
Wood et al. [100]	Isotopic ratio	Multivariate linear regression	670.79 nm	*R*^2^ > 0.9 *RMSE* < 0.09
Hull et al. [101]	Isotopic ratio	Line fitting + univariate calibration curve	670.79 nm	*MAPE* = 13%
Touchet et al. [102]	Isotopic ratio	Line fitting + univariate calibration curve	670.79 nm	*MAPE* = 6%
Gallot-Duval et al. [103]	Isotopic ratio	Line fitting + univariate calibration curve	670.79 nm	*R*^2^ = 0.998 *RMSE* = 6%
Moran et al. [104]	Isotopic ratio	Stacking of multiple ML algorithms	670.79 nm	*RMSE* < 6% *LOD* < 20%
Tran et al. [105]	Isotopic ratio in liquids	Line fitting + univariate calibration curve	670.79 nm	*R*^2^ = 0.998 *RMSE* < 6%
Lee et al. [106]	Liquid	Univariate calibration curve	670.79 nm	*LOD* = 0.8 ppb
Sarkar et al. [107]	Pressurized water reactor	Univariate calibration curve	670.79 nm	*LOD* = 0.01 μg/g
Feng et al. [115]	Tokamak blanket system	CF-LIBS (Er/Li ratio)	Multi-elemental	*LOD* < 20
Ollila et al. [121]	Mars geology	Univariate calibration curve	670.79 nm	*RMSE* = 36 ppm
Payré et al. [122]	Simulated Mars geological samples	Univariate calibration curve	670.79 nm	*RMSE* = 5 ppm
Ytsma et al. [123]	Simulated Mars geological samples	Multivariate linear calibration	Multivariate	*R*^2^ = 0.998 *RMSE* = 27 ppm
Luo et al. [124]	Mars geology	Univariate calibration curve	670.79 nm	*LOD* = 6 ppm *RMSE* = 5 ppm

**Table 5 sensors-25-07689-t005:** Comparison of LIBS technique with other emerging analytical techniques for lithium analysis.

Technique	Principle	Typical Li Application	Main Advantages	Limitations/Challenges	Technology Readiness
LIBS	Laser-induced plasma emission; spectral lines analyzed for elemental quantification	Solid and liquid Li-bearing samples (minerals, brines, black mass)	Fast, reagent-free, portable, minimal waste	Quantitative accuracy limited; matrix effects; weak Li lines	High (industrial prototypes available)
TDLAS	Absorption of tunable diode laser through vapor or plasma to detect Li atomic lines	In situ plasma or gas monitoring (fusion, metallurgy)	High selectivity, real-time measurement	Requires vapor phase; complex optical alignment	Medium (laboratory and pilot systems)
Raman	Inelastic scattering of laser light reveals molecular vibrations	Characterization of Li phases (Li_2_CO_3_, LiOH, battery cathodes)	Non-destructive; chemical speciation possible	Weak Raman signal for Li; fluorescence interference; poor quantification	High (routine for battery materials)
XPS	Photoemission of electrons under X-ray irradiation reveals binding energies	Surface analysis of Li compounds and interphases (battery SEI)	Chemical state information, depth profiling	High vacuum required; expensive; surface-limited	Medium–High (research and industrial R&D)

**Table 6 sensors-25-07689-t006:** Main limitations and challenges in the application of the LIBS technique for lithium analysis.

Category	Main Limitations	Future Challenges and Perspectives
Spectral/Plasma physics	Weak Li emission lines (610.36, 670.8 nm) Self-absorption and line self-reversal above ~0.5 wt% Spectral interferences from Na, K, Ca, Mg Departure from LTE and optically thin plasma assumptions	Improve plasma modeling and self-absorption compensation Develop higher-resolution and dynamically calibrated spectrometers
Analytical/Calibration	Need for matrix-matched standards Large signal variability among solid and liquid matrices Accuracy often limited (10–30% errors) CF-LIBS highly sensitive to plasma and matrix inhomogeneity	Implement hybrid calibration (CF + ML) Develop certified reference materials for geological and recycling matrices
Instrumental	Large performance gap between lab and hand-held systems Sensitivity to alignment, laser energy and gate delay Difficulty in controlling plasma formation on heterogeneous samples	Miniaturization with automated plasma control Multi-laser systems for adaptive ablation
Data and Machine Learning	Overfitting risk due to limited or unbalanced datasets Lack of open benchmark datasets Model transferability limited by matrix effects	Build open-access spectral databases Inter-laboratory validation of ML/DL models Introduce explainable AI approaches for spectral interpretation
Standardization/Interoperability	Absence of harmonized international protocols (ISO, ASTM) Limited cross-laboratory comparability	Develop standardized operating procedures (SOPs) Benchmark LIBS vs. ICP-MS/OES for validation
Industrial and applied context	Limited industrial adoption and field validation Environmental sensitivity of portable devices Lack of robustness testing in real production environments	Integrate LIBS into automated QA/recycling lines Combine LIBS with Raman/XRF sensors (sensor fusion) Demonstrate traceability and sustainability monitoring along the Li supply chain

## Data Availability

Data are contained within the article.

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
