# Peer review of "Laser-Induced Breakdown Spectroscopy Analysis of Lithium: A Comprehensive Review"

_sensors, 2025, doi:10.3390/s25247689_

Round 1
Reviewer 1 Report
Comments and Suggestions for Authors
Authors are well known team in LIBS community. They provided review on Li analysis by LIBS.
I think that lithium electronic structure discussion should be added to the manuscript since Li is a simple element with only 3 electrons so it had simple electronic level structure.
Also I’m confused that authors stated that they found out 225 papers on Li analysis by LIBS but only 106 references were provided in the manuscript. It is not clear for me why 50% of the papers were not reviewed.
I suggest to accept the manuscript after major revision.
Questions
- Table 2. The acronyms like ICP-OES are well known but these acronyms should be decrypt in the text or table’s description. Or provide reference to abbreviation list in the paper. Why atomic emission spectrometry with the spark/arc discharge is missing in the table 2?
- The lithium is rather remarkable element from spectroscopic point of view since its lines 610 and 670 nm had upper level energy of 3.9 and 1.8 eV. So, the optimal choice of analytical line of interest will depend on plasma temperature and electron density. In my opinion it will be convenient to discuss it. Moreover, lithium is a number 3 element in periodic table so it had simple electronic level structure. Please, provide Grotrian diagram for Li and discuss it in details in order to improve the manuscript. I think discussing energy levels and transitions will improve the manuscript quality especially for LIBS newcomer (like PhD students).
- Raman spectroscopy is another powerful technique for on site and in situ analysis. According to the manuscript introduction majority of the Li-containing materials are crystals. These crystals had Raman spectra that can be used for express analysis. Please, discuss Raman spectroscopy application foe Li materials analysis. Authors provided some discussion by combined LIBS and Raman (lines 385-390). However, Raman spectroscopy itself had very nice perspective for Li materials analysis including on site and in situ applications.
- Section 3.3 I recommend to provide figures with the 2D or 3D maps for LIBS signals. It will provide the beauty and power of the LIBS technique for the reader in my opinion.
- Please, provide a new table with the comparing Li analysis sensitivity (limit of detection) and accuracy (RMSECV or alternative). Such information is scattered in the text but should be summarized in the table. It will be convenient to provide the chosen analytical lines for every reference and/or mark chemometric approach utilized for quantitative determination.
- In introduction authors stated that 225 papers have been published on Li analysis by LIBS (…“with keywords “laser AND induced AND breakdown AND spectroscopy AND lithium” reports publications (225 articles, 42 conference papers, 5 reviews, 1 book chapter, and 1 note)…“. However, the manuscript contains only 106 references so more than 50% of the papers have not been presented. Please, explain why you miss so many papers?
Author Response
Authors are well known team in LIBS community. They provided review on Li analysis by LIBS.
I think that lithium electronic structure discussion should be added to the manuscript since Li is a simple element with only 3 electrons so it had simple electronic level structure.
Also I’m confused that authors stated that they found out 225 papers on Li analysis by LIBS but only 106 references were provided in the manuscript. It is not clear for me why 50% of the papers were not reviewed.
I suggest to accept the manuscript after major revision.
Questions
- Table 2. The acronyms like ICP-OES are well known but these acronyms should be decrypt in the text or table’s description. Or provide reference to abbreviation list in the paper.
The description of the acronyms is provided at the end of the manuscript, before the Bibliography.
- Why atomic emission spectrometry with the spark/arc discharge is missing in the table 2?
We thank the reviewer for the suggestion. We have added Spark-OES to the list of techniques in table 2 (now table 3), and in figure 3 (now figure 4).
- The lithium is rather remarkable element from spectroscopic point of view since its lines 610 and 670 nm had upper level energy of 3.9 and 1.8 eV. So, the optimal choice of analytical line of interest will depend on plasma temperature and electron density. In my opinion it will be convenient to discuss it. Moreover, lithium is a number 3 element in periodic table so it had simple electronic level structure. Please, provide Grotrian diagram for Li and discuss it in details in order to improve the manuscript. I think discussing energy levels and transitions will improve the manuscript quality especially for LIBS newcomer (like PhD students).
Thank you very much for the fruitful suggestions. We have added the Grotrian diagram for Li and the relevant transitions in the Introduction.
- Raman spectroscopy is another powerful technique for on site and in situ analysis. According to the manuscript introduction majority of the Li-containing materials are crystals. These crystals had Raman spectra that can be used for express analysis. Please, discuss Raman spectroscopy application for Li materials analysis. Authors provided some discussion by combined LIBS and Raman (lines 385-390). However, Raman spectroscopy itself had very nice perspective for Li materials analysis including on site and in situ applications.
We agree with the reviewer that Raman spectroscopy has great potential for Li materials analysis, despite its limitations in quantitative analysis. We have added a paragraph about the use of Raman spectroscopy in the discussion of Table 4 (now Table 5).
- Section 3.3 I recommend to provide figures with the 2D or 3D maps for LIBS signals. It will provide the beauty and power of the LIBS technique for the reader in my opinion.
Following the reviewers’ recommendation, we added some additional figures to show the potential of LIBS in lithium analysis.
- Please, provide a new table with the comparing Li analysis sensitivity (limit of detection) and accuracy (RMSECV or alternative). Such information is scattered in the text but should be summarized in the table. It will be convenient to provide the chosen analytical lines for every reference and/or mark chemometric approach utilized for quantitative determination.
We added a new table summarizing the analytical figures of merit of the results reported in the text for lithium quantitative determination and the analytical method used by the authors.
- In introduction authors stated that 225 papers have been published on Li analysis by LIBS (…“with keywords “laser AND induced AND breakdown AND spectroscopy AND lithium” reports publications (225 articles, 42 conference papers, 5 reviews, 1 book chapter, and 1 note)…“. However, the manuscript contains only 106 references so more than 50% of the papers have not been presented. Please, explain why you miss so many papers?
We understand that this discrepancy may appear strange. However, the number of papers obtained from Scopus was reported for statistical analysis, but not all papers corresponding to the selected keywords had lithium as a specific research target. We therefore selected and discussed only those papers specifically focused on lithium analysis and, among these, the most relevant to the topics under discussion.
Reviewer 2 Report
Comments and Suggestions for Authors
The authors reported the Laser-induced Breakdown Spectroscopy analysis of lithium: a comprehensive review. The review article is dense with very long paragraphs and lacks a graphical presentation. Also, many cited references do not match the authors’ writing. Moreover, several major aspects, as pointed out below, are missing in the manuscript. Therefore, I suggest that the current version of the manuscript is not suitable for publication.
- There are several sections where the explanations are too dense, especially in 3.1 Rocks, 3.2 Brines, 3.3 Batteries, and 3.4 Other Li-rich materials. Authors should add symmetric data among those methods and include an experimental figure highlighting the core research of the previous publication.
- Authors need to add more diagrams and figures to make it easier for readers to understand. Please highlight critical experimental results and technical issues.
- Authors need to add references to support Table 2. I have checked Reference 8 listed in Table 2, and it does not support the table. Authors should also add proper references for Table 3.
- There are many spelling errors, such as “Canda” to “Canada”. Please check the entire manuscript and correct all spelling mistakes.
- There are many minor grammatical errors that the authors need to fix.
- The manuscript organization and logical flow are poor, and the number of figures is insufficient.
- According to the Abstract and the manuscript organization, many aspects are missing. For example, the authors mention in the abstract: “It also discusses emerging applications in closed-loop recycling and highlights future prospects related to the integration of LIBS with artificial intelligence and machine learning to enhance analytical accuracy and material classification.” However, the manuscript does not explain these points well. Authors need to focus more on the future prospects of LIBS, including artificial intelligence and machine learning.

Author Response
The authors reported the Laser-induced Breakdown Spectroscopy analysis of lithium: a comprehensive review. The review article is dense with very long paragraphs and lacks a graphical presentation. Also, many cited references do not match the authors’ writing. Moreover, several major aspects, as pointed out below, are missing in the manuscript. Therefore, I suggest that the current version of the manuscript is not suitable for publication.
- There are several sections where the explanations are too dense, especially in 3.1 Rocks, 3.2 Brines, 3.3 Batteries, and 3.4 Other Li-rich materials. Authors should add symmetric data among those methods and include an experimental figure highlighting the core research of the previous publication.
Following the reviewer’s suggestion, we have expanded the introductory part of the relevant sections to outline the advantages of using LIBS in the corresponding application context. We believe that the flow of the discussion is now easier to follow, as requested by the reviewer.
- Authors need to add more diagrams and figures to make it easier for readers to understand. Please highlight critical experimental results and technical issues.
We thank the reviewer for their fruitful suggestions. We have tried to improve the readability of the manuscript by adding new figures and diagrams, as also suggested by reviewer #1.
- Authors need to add references to support Table 2. I have checked Reference 8 listed in Table 2, and it does not support the table. Authors should also add proper references for Table 3.
We have reformulated the discussion of the main analytical techniques used for lithium analysis, adding references that describe each technique and their analytical figures of merit. As for Table 3, we decided to remove it since it did not provide information specifically relevant to lithium analysis.
- There are many spelling errors, such as “Canda” to “Canada”. Please check the entire manuscript and correct all spelling mistakes.
- There are many minor grammatical errors that the authors need to fix.
We have revised carefully the manuscript to correct spelling and grammatical errors.
- The manuscript organization and logical flow are poor, and the number of figures is insufficient.
Following the reviewers’ suggestion, we have tried to improve the logical flow of the manuscript and added more figures to support the discussion.
- According to the Abstract and the manuscript organization, many aspects are missing. For example, the authors mention in the abstract: “It also discusses emerging applications in closed-loop recycling and highlights future prospects related to the integration of LIBS with artificial intelligence and machine learning to enhance analytical accuracy and material classification.” However, the manuscript does not explain these points well. Authors need to focus more on the future prospects of LIBS, including artificial intelligence and machine learning.
We have added a new section specifically devoted to the discussion of the analytical methods used for quantitative lithium analysis. In this section, we have added a new Table listing the univariate/multivariate and linear/non-linear techniques used, also following the suggestion of Reviewer #1.
Round 2
Reviewer 1 Report
Comments and Suggestions for Authors
The manusctipt can be accepted in current form.
Reviewer 2 Report
Comments and Suggestions for Authors
The revised manuscript is well presented and clearly describes all the questions.
The manuscript can be accepted for publication.